# Comprehensive Genome-Wide Analysis and Expression Pattern Profiling of the *SlHVA22* Gene Family Unravels Their Likely Involvement in the Abiotic Stress Adaptation of Tomato

**DOI:** 10.3390/ijms232012222

**Published:** 2022-10-13

**Authors:** Antt Htet Wai, Muhammad Waseem, Lae-Hyeon Cho, Sang-Tae Kim, Do-jin Lee, Chang-Kil Kim, Mi-Young Chung

**Affiliations:** 1Department of Agricultural Education, Sunchon National University, 413 Jungangno, Suncheon 57922, Korea; 2Department of Biology, Yangon University of Education, Kamayut Township, Yangon 11041, Myanmar; 3Department of Botany, University of Narowal, Narowal 51600, Pakistan; 4Department of Plant Bioscience, College of Natural Resources and Life Science, Pusan National University, Miryang-si 50463, Korea; 5Department of Medical and Biological Sciences, The Catholic University of Korea, Bucheon 14662, Korea; 6Department of Horticulture, Kyungpook National University, Daegu 41566, Korea

**Keywords:** *HVA22* genes, *Solanum lycopersicum*, stress response, subcellular localization, gene co-expression network, 3D structure, expression analysis

## Abstract

HVA22 family proteins with a conserved TB2/DP1/HVA22 domain are ubiquitous in eukaryotes. *HVA22* family genes have been identified in a variety of plant species. However, there has been no comprehensive genome-wide analysis of *HVA22* family genes in tomato (*Solanum lycopersicum* L.). Here, we identified 15 non-redundant *SlHVA22* genes with three segmentally duplicated gene pairs on 8 of the 12 tomato chromosomes. The predicted three-dimensional (3D) models and gene ontology (GO) annotations of SlHVA22 proteins pointed to their putative transporter activity and ability to bind to diverse ligands. The co-expression of *SlHVA22* genes with various genes implicated in multiple metabolic pathways and the localization of SlHVA22-GFP fused proteins to the endoplasmic reticulum suggested that they might have a variety of biological functions, including vesicular transport in stressed cells. Comprehensive expression analysis revealed that *SlHVA22* genes were differentially expressed in various organs and in response to abiotic stress conditions. The predominant expression of *SlHVA22i* at the ripening stage and that of *SlHVA22g*, *SlHVA22k*, and *SlHVA22l* in fruits at most developmental stages suggested their probable involvement in tomato fruit development and ripening. Moreover, the transcript expression of most tomato *HVA22* genes, particularly *SlHVA22b*, *SlHVA22i*, *SlHVA22k*, *SlHVA22l*, *SlHVA22m*, and *SlHVA22n*, was affected by abscisic acid (ABA) and diverse abiotic stress treatments, indicating the likely involvement of these genes in tomato abiotic stress responses in an ABA-dependent manner. Overall, our findings provide a foundation to better understand the structures and functional roles of *SlHVA22* genes, many of which might be useful to improve the abiotic stress tolerance and fruit quality of tomato through marker-assisted backcrossing or transgenic approaches.

## 1. Introduction

TB2/DP1/HVA22 family proteins are prevalent in eukaryotes, such as plants, fungi, animals, and protozoa, but have not yet been identified in prokaryotes [1,2,3]. The plant HVA22 protein shares a high sequence similarity with the human TB2/DP1 protein. HVA22 was first identified as an abscisic acid (ABA)- and stress-induced gene from barley (*Hordeum vulgare*) [4]. Transmembrane domains that are critical for the proper localization and functions of HVA22 proteins, such as regulation of programmed cell death in aleurone cells and vesicular trafficking in stressed cells, are prevalent in the TB2/DP1/HVA22 domain regions of HVA22 family members [1,5,6]. To date, over 355 HVA22 homologous proteins have been identified in eukaryotes, and the TB2/DP1/HVA22 domain is highly conserved in all of them [1].

Yop1p, identified in yeast, is homologous to HVA22 and capable of interacting with diverse proteins, including the endoplasmic reticulum (ER) resident protein Rtn4/NogoA, to regulate protein interactions and ER functions [7,8]. The interaction of Yop1p and Sey1p during vesicular trafficking was elucidated by the agglomeration of transport vesicles and the reduction in invertase secretion in the yeast *yop1/sey1* double mutant [5]. mRNA accumulation of *Yop1* is induced under salt treatment, and *yop1* mutants also show sensitivity to mild temperature stress compared with the control [3,5]. *HVA22* induced by ABA regulates seed germination and seedling growth by modulating vesicular trafficking in barley aleurone cells [1,5]. Plant *HVA22* homologous genes responsive to environmental stresses harbor *cis*-regulatory elements associated with stress-related hormones and stress tolerance in their promoters [6,9,10,11]. The interaction between the *cis* elements in the *HVA22* promoter and ABA- and stress-related genes has been verified in several plant species [12,13,14,15].

Plants, as sessile organisms, are occasionally exposed to unfavorable environmental conditions during growth and development. Abiotic stressors, including heat, cold, drought, and salinity, have a profound effect on the growth and development of plants, resulting in significant crop yield losses worldwide [16]. Diverse abiotic stresses differentially regulate the expression of *HVA22* homologs in various plant species [3,17,18]. The *HVA22* family members identified in *Arabidopsis* are differentially expressed on exposure to the plant stress hormone ABA and abiotic stresses [17]. The introgression of stress-related genes under the control of the *HVA22* promoter as a stress-inducible promoter in rice enhances drought tolerance of the resulting rice transgenic lines compared with wild-type (WT) plants [19]. Six *CcHVA22* homologs have been identified in two citrus species (*Citrus sinensis* and *Citrus clementina*), and *CcHVA22d*-overexpressing tobacco transgenic lines display tolerance to dehydration and oxidative stress [6]. The *HVA22* homolog from halophytic cordgrass (*Spartina alterniflora*) displays varied transcript levels on exposure to different salinity levels [18]. Likewise, the tomato *HVA22* homolog (referred to as *SlHVA22n* here) is significantly induced by salt treatment in the tomato (*Solanum lycopersicum* L.) cultivar Micro-Tom. It is also upregulated in *LeERF1*- and *LeERF2*-overexpressing tomato plants that are tolerant to salt stress compared with WT plants [20].

Tomato is a commercially valuable fruit crop as well as a fleshy fruit model plant used to study the mechanisms of fruit ripening. Different unfavorable environmental conditions significantly reduce the productivity and fruit quality of tomato [16]. Therefore, extensive research has been conducted to better understand the molecular pathways controlling fruit development and ripening in tomato and to maximize fruit yield and quality under environmental stresses. In this study, we analyzed the expression profiles of the tomato *HVA22* gene family at different developmental stages, including five fruit developmental stages, and under diverse abiotic stress treatments. We conducted systematic and comprehensive genome-wide characterization as well as phylogenetic and gene co-expression network analyses. Finally, we determined the subcellular localization of the *SlHVA22* genes. Our results will provide a solid foundation for future functional elucidation of potential tomato *HVA22* genes related to development and abiotic stress tolerance of tomato.

## 2. Results

### 2.1. In Silico Identification and Phylogenetic Analysis of Tomato HVA22 Family Proteins

We identified 15 *HVA22* tomato genes and designated them as *HVA22a*–*HVA22o* based on their chromosomal locations. The characteristics of *SlHVA22* genes and their corresponding proteins are presented in Appendix A. The protein sequence identity ranged from 10% to 73% among SlHVA22 proteins, and the proteins that were clustered in the same phylogenetic groups had higher sequence identities than those from different groups (Appendix A). The phylogenetic analysis classified diverse plant HVA22 family proteins into two major clades, which were further divided into four groups according to their phylogenetic relationship (Figure 1).

The 15 tomato HVA22s were distributed in all four groups, with the largest number of proteins (SlHVA22d, SlHVA22f, SlHVA22h, SlHVA22j, and SlHVA22k) in group I and the smallest number (SlHVA22c and SlHVA22l) in group III. The other two groups had four proteins: group II contained SlHVA22b, SlHVA22e, SlHVA22i, and SlHVA22n, and group IV contained SlHVA22a, SlHVA22g, SlHVA22m, and SlHVA22o. All SlHVA22 members preferentially clustered with their homologs from potato, which is an evolutionarily closely related species, except SlHVA22j, which was in the same cluster with the homologs from rice and sorghum. The HVA22 members from the basal angiosperm *Amborella trichopoda* were distributed in all four groups and clustered with their homologs from dicotyledonous species, except for AmtrHVA22c, which paired with the monocot homologs in group I. In contrast, the corresponding proteins from the moss *Physcomitrella patens* were only in groups II and III. The moss homologs were distributed evenly in both groups, with six in group II and seven in group III, while those of the unicellular green alga were present solely in group III.

### 2.2. Gene Structure, Conserved Motif and Domain Analysis of Tomato HVA22 Genes

Analysis of the exon–intron organization revealed structural divergence in the tomato *HVA22* gene family (Appendix A). The number of exons in the *SlHVA22* members varied from two to eight, with a mean of five. However, the exon–intron composition of most *HVA22* genes in the same phylogeny group was identical or similar, highlighting the structural similarity within phylogenetically closely related members. Likewise, the majority of the SlHVA22 homologs clustered together in the same phylogenetic clades contained similar conserved motif organization, indicating structural similarity among phylogenetically closely related members (Appendix A). We also identified several motifs unique to some HVA22 members, such as motif 6 for SlHVA22g and SlHVA22o, motif 10 for SlHVA22c and SlHVA22l, and motif 7 for SlHVA22d and SlHVA22h. The conserved motifs identified were similar in the homologous proteins clustered in the same phylogenetic groups from all three plant species (*Arabidopsis*, tomato, and rice), implying they are highly conserved in dicots and monocots. Motifs 1 and 2 were located in the TB2/DP1/HVA22 domain and were present in all HVA22 homologous proteins from *Arabidopsis*, tomato, and rice, except for OsHVA22b, indicating that these important motifs were evolutionarily conserved in both dicot and monocot plants. The domain architecture of *SlHVA22* genes was diverse, varying from the presence of the single TB2/DP1/HVA22 domain to the composition of other additional domains such as transmembrane domains, ZnF_U1 domains, ZnF_C2H2 domains, and RVT_3 domains (Figure 2). The TB2/DP1/HV- -A22 domains with a length of 77–79 aa in tomato HVA22 proteins were highly conserved and started with the conserved Pro residue (except in SlHVA22e that started with Cys residue). The transmembrane domains (TMDs) prevalent in the TB2/DP1/HVA22 domain regions were also conserved in the majority of SlHVA22 family proteins, with the exception of SlHVA22c and SlHVA22f. Other types of domains (ZnF_U1, ZnF_C2H2, and RVT_3) were only identified in SlHVA22f and SlHVA22k. Analysis of the multiple sequence alignment of the TB2/DP1/HVA22 domain of SlHVA22 family members with that of the human homolog TB2/DP1 protein revealed the evolutionary conservation of HVA22 family proteins from a human and a dicotyledonous plant, including the conservation of the casein kinase II (CKII) phosphorylation sites ([S/T]XX[D/E]) in the third α-helices of the domain (Figure 3) [3].

### 2.3. Chromosomal Position, Gene Duplication, and Microsynteny Analysis of SlHVA22 Genes

*SlHVA22* genes were unevenly situated on 8 of the 12 chromosomes (Chr) of tomato, and most of them were located close to the distal portions of the chromosomes (Appendix A). Five of these chromosomes (Chr 1, 5, 6, 11, and 12) carried only one gene each, while Chr 3 and 4 harbored three, and Chr 10 harbored four genes. Three segmentally duplicated gene pairs (*SlHVA22a*/*SlHVA22m*, *SlHVA22e*/*SlHVA22n*, and *SlHVA22g*/*SlHVA22o*) were predicted in tomato, two of which belonged to phylogenetic group IV, whereas the remaining pair belonged to group II (Figure 1 and Appendix A). The duplicated genes of each pair were located on different chromosomes, one of which carried only a single *SlHVA22* gene, while the other had three or four *SlHVA22* genes. Tandem duplication events were not detected because the genes were not mapped within the 100-kb region on the same chromosome. The Ka/Ks ratio of all the duplicated gene pairs in the tomato *HVA22* gene family was less than one, revealing that these genes have been influenced by intense purifying selection over the course of evolution (Table 1). We estimated that these duplicated gene pairs diverged 13.77–25.2 million years ago. We conducted a comparative microsynteny analysis to determine *HVA22* orthologous gene pairs among *Arabidopsis*, tomato, and rice to discern the evolutionary correlation across their genomes (Figure 4). We identified four orthologous gene pairs between *Arabidopsis* and tomato, whereas there were no orthologous gene pairs between *Arabidopsis* and rice or tomato and rice.

### 2.4. Prediction of Cis-Regulatory Elements, microRNA (miRNA) Target Sites and Phosphorylation Sites

Hormonal regulation of gene expression is also vital in the control of stress responses in plants [21]. We identified many *cis*-regulatory elements related to phytohormones and abiotic stress responses in the promoter regions of tomato *HVA22* genes. These included ABA-responsive elements; TGACG and CGTCA motifs (with roles in the jasmonic acid response); TCA elements (related to the salicylic acid response); P-box, GARE and TATC elements associated with the gibberellic acid response; auxin response-related TGA-box and TGA elements; MYB-binding site and MYB elements responsive to drought; low temperature-responsive elements with roles in low temperature and hypersalinity stress and defense; and stress response-related TC-rich repeats (Appendix A). miRNAs function in the regulation of biotic and abiotic stress tolerance in plants by targeting a wide range of stress-related genes [22,23]. We identified many miRNA target sites related to abiotic stress tolerance in tomato in most *HVA22* genes, except *SlHVA22e*, *SlHVA22h*, and *SlHVA22i* (Appendix A). Nineteen miRNA target sites involved in abiotic stress tolerance in tomato were predicted in 11 *SlHVA22* genes. Post-translational regulation of stress-related proteins via phosphorylation is common and important in plant stress responses. We identified numerous phosphorylation sites, including the sites targeted by CKII and several N-glycosylation sites, in tomato HVA22 proteins (Appendix A).

### 2.5. Comparative Modelling of Tomato HVA22 Proteins

The predicted 3D models indicated that the TB2/DP1/HVA22 domain (77–79 aa) was present as a 3D frame comprised mainly of α-helixes in all tomato HVA22 proteins, most of which were only composed of α-helixes and coils, while a few (SlHVA22f, SlHVA22k, and SlHVA22o) also included β-strands (Figure 5). The number of secondary structural components of tomato HVA22 proteins was 4–22 for α-helixes, 0–9 for β-strands, and 5–27 for coils (Appendix A). The number of secondary structural components was highly conserved in most tomato proteins, with only 4 or 6 α-helixes and 5 or 7 coils, whereas only two proteins had more α-helixes and coils (13 and 22 for HVA22f, respectively, and 22 and 27 for SlHVA22k, respectively). Only 3 of 15 predicted models contained β-strands (2 for HVA22o, 4 for HVA22k, and 9 for SlHVA22f). The C-scores and TM-scores along with other parameters of all the predicted models that were within a reasonable range are described in Appendix A to indicate the validity of the constructed models [24]. We identified the putative ligand-binding sites capable of interacting with diverse molecules in all predicted models. We predicted the molecular functions of SlHVA22 family members, including binding ability to a variety of ligands, transporter activity, and transferase activity, based on gene ontology (GO) terms using the I-TASSER server (Appendix A).

### 2.6. Gene Co-Expression Network Analysis

We determined the co-expression profiles of *SlHVA22* genes using RNA sequencing data with weighted gene co-expression network analysis and showed that 263 genes were co-expressed with *SlHVA22* genes (Figure 6). The hub genes (*SlHVA22a*, *SlHVA2g*, *SlHVA*- *-22k*, and *SlHVA22l*) were co-expressed with 111, 107, 23, and 22 genes, respectively. In the GO and Kyoto Encyclopedia of Genes and Genomes enrichment results, certain genes present in the co-expression network were not annotated with any biological process, but most genes were engaged in a variety of biological pathways, including environmental information processing, signaling and cellular processes, nucleic acid binding, oxidative phosphorylation, starch and sucrose metabolism, transport and catabolism, biosynthesis of secondary metabolites, membrane transport, lipid metabolism, metabolism of amino acids, mRNA metabolism, and phosphorus metabolic processes (Figure 6, Appendix A).

Genes related to development and abiotic stress responses of tomato were identified in the co-expression network. Numerous genes responsive to diverse abiotic stresses and/or expressed in roots were co-expressed with *SlHVA22a*. We also identified several genes that were responsive to abiotic stresses and/or expressed in fruits in the co-expression networks of *SlHVA22g*, *SlHVA22k*, and *SlHVA22l*.

### 2.7. Subcellular Localization of SlHVA22 Proteins

The predicted localization analysis showed that tomato HVA22 proteins were localized to various parts of the cell, including the ER, chloroplasts, and nucleus (Appendix A). Further determination of subcellular locations of SlHVA22 proteins through the expression of these proteins fused with GFP in the rice protoplasts indicated that SlHVA22a, SlHVA22f, and SlHVA22n were predominantly localized in the ER (Figure 7).

### 2.8. Expression Profiling of Tomato HVA22 Genes in Different Organs

The expression profiles of SlHVA22 genes in tomato organs revealed differential expression patterns in all tested organs (Figure 8). We determined expression levels in various organs relative to those in leaves (control). The duplicated gene pair, *SlHVA22a* and *SlHVA22m*, had high relative expression levels in roots (4- and >5-fold higher than the control, respectively). *SlHVA22m* was the only gene whose expression was not detected in young (1-cm) or immature (IM) fruit. *SlHVA22b*, *SlHVA22h*, and *SlHVA22j* had higher relative expression in stems than in other vegetative and reproductive organs (>20-, 14-, and 2-fold higher, respectively, than in the control), but the peak expression of *SlHVA22h* in stems was similar to that in leaves and flowers. *SlHVA22d* is the only gene whose expression was highest in leaves, followed by mature green (MG) fruit, IM fruit, and fruit 5 d after the breaker stage (B5), respectively. *SlHVA22f* and *SlHVA22o* had higher relative expression in flowers compared to in other organs (>2- and ~2-fold higher, respectively, than in the control). *SlHVA22c* had a ~2-, ~1.9- and 1.7-fold higher expression levels in IM fruit, stems, and B5 fruit relative to the control, respectively. *SlHVA22k* had the highest expression level in MG fruit (2.6-fold higher than in the control), followed by IM fruit, B5 fruit, and fruit at the breaker stage (B), respectively. *SlHVA22n* was predominantly expressed in MG fruit (~4.7-fold higher expression than in the control); it also showed a >4-fold higher expression in flowers and roots compared with the control. *SlHVA22i* was preferentially expressed in B5 fruit (>290-fold higher expression than the control) and in B fruit (>80-fold higher than the control). *SlHVA22g* exhibited a ~2-fold higher expression level in B5 fruit than in the control and was also highly expressed in B and IM fruit. *SlHVA22l*, whose expression peaked in B5 fruit (when it was 3.5-fold higher than that in control), displayed higher expression in the cell expansion (IM and MG) and ripening (B and B5) stages of tomato compared with other organs.

### 2.9. Expression Analysis of SlHVA22 Genes in Response to Abiotic Stresses and Phytohormone Treatment

We determined the transcript profiles of tomato *HVA22* gene family members in leaf samples exposed to different abiotic stresses (cold, heat, drought, and salt) and phytohormone (ABA) treatment via a qRT-PCR assay. Many *SlHVA22* genes were differentially expressed in response to these stress treatments (Figure 9A–E).

Nine genes (*SlHVA22c*, *SlHVA22d*, *SlHVA22f*, *SlHVA22g*, *SlHVA22h*, *SlHVA22k*, *SlHVA22l*, *SlHVA22m*, and *SlHVA22o*) were significantly downregulated by cold stress (1.4- to 12-fold lower expression compared with that in the control [0 h sample]). In contrast, five genes (*SlHVA22a*, *SlHVA22b*, *SlHVA22e*, *SlHVA22i*, and *SlHVA22n*) were significantly upregulated by 1.4- to 7.8-fold under cold stress. Of these, *SlHVA22i* and *SlHVA22n* were upregulated by >6- and 7-fold, respectively, compared with the control. *SlHVA22j* was the only gene not responsive to cold treatment (Figure 9A).

All tomato *HVA22* genes displayed heat stress responses. Eleven genes (*SlHVA22b*, *SlHVA22c*, *SlHVA22d*, *SlHVA22e*, *SlHVA22g*, *SlHVA22i*, *SlHVA22j*, *SlHVA22k*, *SlHVA22l*, *SlHVA22m*, and *SlHVA22o*) were significantly upregulated by heat stress in comparison with the control (by 1.9- to 16.5-fold). In particular, the transcript levels of *SlHVA22b*, *SlHVA22g*, *SlHVA22l*, and *SlHVA22o* were upregulated by >3-fold and those of *SlHVA22i* by >16-fold in response to heat treatment. In contrast, four genes (*SlHVA22a*, *SlHVA22f*, *SlHVA22h*, and *SlHVA22n*) were downregulated (by 1.9- to 2.5-fold) under heat treatment (Figure 9B). 

Drought stress also affected the expression levels of *SlHVA22* genes. *SlHVA22i* expression was strikingly upregulated by 40- to 126-fold at 3–24 h of drought treatment in comparison to the control. Likewise, *SlHVA22k*, *SlHVA22l*, and *SlHVA22n* were markedly upregulated by 2.5- to 8.5-fold under drought stress. On the other hand, *SlHVA22e*, *SlHVA22f*, *SlHVA22j*, *SlHVA22h*, *SlHVA22m*, and *SlHV**A22o* were significantly downregulated by 2.2- to 17-fold after exposure to drought treatment (Figure 9C).

Salt treatment either markedly up or downregulated most *SlHVA22* genes in comparison to the control, except for *SlHVA22j*, which was only minimally up- or downregulated (by ~1.3-fold). Eleven of the fifteen genes (*SlHVA22a*, *SlHVA22b*, *SlHVA22c*, *SlHVA22e*, *SlHVA22f*, *SlHVA22g*, *SlHVA22i*, *SlHVA22k*, *SlHVA22l*, *SlHVA22m*, and *SlHVA22n*) were significantly upregulated by 1.6- to 63-fold under cold stress in comparison with the control. Among the upregulated genes, the expression of *SlHVA22m*, *SlHVA22i*, *SlHVA22b*, *SlHVA22n*, and *SlHVA22l* was upregulated by 63-, 16-, 9-, 6-, and 3-fold compared with the control, respectively, under exposure to saline conditions. However, a few genes (*SlHVA22h*, *SlHVA22j*, and *SlHVA22o*) were downregulated by 1.8- to 3.3-fold under salt stress compared with the control (Figure 9D).

The transcription of many tomato *HVA22* genes, apart from *SlHVA22d*, was affected by ABA stress hormone treatment. *SlHVA22o* was downregulated (by ~1.3-fold) at all time points, and *SlHVA22j* was downregulated (by 1.4-fold) at the last time point of ABA treatment compared with the control. In contrast, many genes (*SlHVA22a*, *SlHVA22c*, *SlHVA22e*, *SlHVA22g*, *SlHVA22h*, *SlHVA22k*, and *SlHVA22l*) were upregulated by 1.4- to 1.8-fold under ABA treatment compared with the control. *SlHVA22i* was upregulated by 1.9- to >7-fold under ABA treatment at all time points compared with the control. *SlHVA22f* was significantly upregulated by 2.2- to 4.7-fold under exposure to ABA at the early stages of treatment. Similarly, the expression of *SlHVA22b* was upregulated by 1.8- to 2-fold at the early stages of ABA treatment, while that of *SlHVA22b* peaked at 9 h after treatment, when it was 3-fold higher than that in the control (Figure 9E).

## 3. Discussion

TB2/DP1/HVA22 family proteins have been identified in eukaryotes but not in prokaryotes [1,2,3], indicating that they might have evolved after the divergence of prokaryotes and eukaryotes, playing a vital role in eukaryote evolution. Plant *HVA22* genes form a multigene family in several plant species and play a role in plant development and adaptation to various abiotic stresses [1,5,6,17,18]. Genome-wide characterization of *HVA22* genes and their roles in plant abiotic stress responses have been documented in multiple plant species, but not yet in any solanaceous crops. In this study, we identified and comprehensively characterized 15 non-redundant *SlHVA22* genes at a genome-wide scale. As in other flowering plant species, the existence of *HVA22* genes as a multiple gene family in tomato (Figure 1) suggested their important biological role in this model fruit crop.

We performed a phylogenetic analysis to explore the evolutionary relationship among HVA22 homologs from diverse species in the plant kingdom. The corresponding proteins from the single-celled green alga *Chlamydomonas reinhardtii* only occurred in phylogenetic group III, revealing that the HVA22 proteins from phylogenetic group III were more primitive than proteins in other groups (I, II, and IV) and may have evolved from the common ancestor of chlorophytes and streptophytes, which diverged over one billion years ago [25]. The homologous proteins from streptophytes were clustered in phylogenetic group II, hinting at the emergence of group II members from the common progenitor existing prior to the split between bryophytes and angiosperms. The prevalence of only the HVA22 homologs from angiosperms in groups I and VI indicated that the homologs from these phylogenetic groups may have evolved before the divergence of monocots and dicots (~200 million years ago) [26].

All *HVA22* genes in the *Citrus* species contain TMDs [6]; however, the minority of *SlHVA22* genes were devoid of TMDs (Figure 2), suggesting the probable role of this domain in the structural and functional divergence of *HVA22* genes in tomato. In addition, *SlHVA22* genes contained other types of domains, such as zinc-finger domains identified in DNA- and RNA-binding proteins (the C2H2-type zinc-finger domain and the U1-type zinc-finger domain) and RVT_3 domains in a few tomato genes. These domains may have also enhanced the diversification of the tomato *HVA22* gene family. The conserved motifs, exons, and introns identified in *SlHVA22* gene family members were arranged in a similar manner across evolutionarily closely related *SlHVA22* genes but in a dissimilar arrangement from those of different phylogenetic groups (Appendix A). This could explain the functional similarity among evolutionarily closely related *SlHVA22* genes and the functional dissimilarity across divergent ones over the course of evolution.

Three segmentally duplicated gene pairs (*SlHVA22a*/*SlHVA22m*, *SlHVA22e*/*SlHVA22n*, and *SlHVA22g*/*SlHVA22o*) were predicted in tomato (Appendix A, Table 1). Therefore, *HVA22* family members might have evolved from an original set of 12 progenitor genes in tomato. The duplicated genes of each pair resided on different chromosomes, one of which contained only one *HVA22* gene, whereas the others had three or four *HVA22* genes (Appendix A), suggesting that gene duplication increased not only the number of *HVA22* genes in the tomato genome but also the number of chromosomes carrying them in tomato. This result indicates that more chromosomes in tomato may have needed to harbor *HVA22* genes during evolution to boost important biological functions in tomato cells, such as adaptation to unfavorable environmental conditions. 

A comparative microsyntenic map constructed to explore the evolutionary relationship among *HVA22* orthologs from *Arabidopsis*, tomato, and rice revealed four pairs of orthologous genes between *Arabidopsis* and tomato but none between rice and *Arabidopsis* or tomato (Figure 4). These results are well correlated with the closer evolutionary connection of tomato with *Arabidopsis* than with rice, and they also suggest that four *SlHVA22* genes might have derived from *Arabidopsis* during species divergence.

*Cis*-acting elements in the promoter sequences of genes can act like circuit breakers to switch the transcription of their genes on and off upon exposure to different environmental stimuli [27,28]. The presence of several *cis*-regulatory elements related to hormonal and abiotic stress responses upstream of tomato *HVA22* genes highlighted their probable roles in tomato abiotic stress tolerance (Appendix A). This result is corroborated by the prevalence of hormonal and stress-related *cis* elements, which can interact with diverse *trans*-acting genes, including stress-related transcription factors, in the promoter regions of stress-induced *HVA22* genes in other plant species [1,3,6,13,17].

To determine whether the abiotic stress responses of *SlHVA22* genes could also be related to miRNAs, we analyzed the miRNA target sites in *SlHVA22* genes. Eleven out of fifteen *SlHVA22* gene family members were targeted by tomato miRNAs that regulate abiotic stress tolerance in tomato, such as *Sly-MIR159b*, *Sly-MIR166c-5p*, *Sly-MIR1917*, *Sly-MIR395a*, *Sly-MIR396a*, *Sly-MIR396a-5p*, *Sly-MIR482a*, *Sly-MIR5302a*, *Sly-MIR5303*, *Sly-MIR6023*, *Sly-MIR6024*, *Sly-MIR9470-5p*, *Sly-MIR9474-5p*, *Sly-MIR9479*, and *sly-MIR9479-3p* [29,30,31,32] (Appendix A). This indicates that many *SlHVA22* genes might be linked to the post-transcriptional regulation of miRNAs in tomato abiotic stress tolerance. 

Protein phosphorylation, a crucial post-translational modification regulated by kinases and phosphatases, is crucial in signaling pathways and stress responses in plants [33,34]. Prediction of phosphorylation sites using the NetPhos 3.1 Server revealed that many putative phosphorylation sites were prevalent in all tomato HVA22 proteins (Appendix A), which is consistent with the distribution of phosphorylation sites in HVA22 homologs in other species [3,4,6].

The 3D structure of a protein can provide useful clues to predict its possible interaction with other molecules and its biological functions. Thus, we analyzed 3D models of tomato HVA22 proteins to gain a better understanding of their molecular structural conformations and putative functions. Most HVA22 proteins had similar numbers of α-helixes, β-strands, and coils (Appendix A), suggesting structural conservation in most HVA22 family members during evolution. SlHVA22 proteins were predicted to harbor ligand-binding sites that interact with various molecules, such as ions, intracellular messengers, or receptor molecules, to initiate a change in cell function. SlHVA22 proteins were also predicted to have various molecular functions, including the ability to bind to a variety of ligands, transporter activity, and transferase activity based on the GO terms (Figure 5, Appendix A). These results indicate that SlHVA22 proteins might have several important biological functions in tomato.

The determination of protein subcellular localization can indicate putative functions. Subcellular localization analysis revealed that tomato HVA22 proteins were predominantly localized to the ER (Figure 7). This finding was consistent with the localization of HVA22 homologs from rice, barley, and yeast in the ER or the ER and Golgi apparatus, suggesting that HVA22 homologs from diverse species might have a conserved function, such as vesicular trafficking in abiotic stress responses [1,8,35].

A qRT-PCR assay revealed varied expression patterns of *SlHVA22* genes in the different organs tested, suggesting that they might have distinct regulatory functions in the growth and development of tomato (Figure 8). Of the 15 tomato *HVA22* genes, eight genes (*SlHVA22c*, *SlHVA22f*, *SlHVA22g*, *SlHVA22i*, *SlHVA22k*, *SlHVA22l*, *SlHVA22n*, and *SlHVA22o*) had high transcript levels in reproductive organs (flowers or fruits), whereas seven genes (*SlHVA22a*, *SlHVA22b*, *SlHVA22d*, *SlHVA22e*, *SlHVA22h*, *SlHVA22j*, and *SlHVA22m*) had high transcript levels in vegetative organs, such as leaves, roots, or stems, hinting at their preferential roles in these organs and developmental stages in tomato.

The expression of *SlHVA22d* was the highest in leaves, IM fruit, MG fruit, and B5 fruit, pointing to its possible role in the growth of leaves and fruit. The roots play a vital role in the uptake of water and minerals for the growth and development of plants. The mRNA transcript levels of *SlHVA22a* and *SlHVA22m* were highest in roots, suggesting their probable involvement in root growth and uptake of nutrients and water. The stem is the main organ that provides mechanical strength to the aerial parts of the plant and transports water and nutrients to promote plant growth and development under normal and unfavorable environmental conditions. The higher mRNA transcript accumulation of *SlHVA22b*, *SlHVA22e*, *SlHVA22h*, and *SlHVA22j* in stems compared to in other organs hints at their likely role in stem development, long-range translocation of water and minerals, as well as stress adaptation (Figure 8**)**.

Of the eight *SlHVA22* genes that were highly expressed in reproductive organs, two genes (*SlHVA22f* and *SlHVA22o*) were predominantly expressed in flowers, suggesting their probable role in flower development. Tomato, as a model fleshy fruit crop, has been widely studied to understand the regulatory mechanisms governing the growth and ripening of climacteric fruits [36,37,38]. The expression of *SlHVA22k* and *SlHVA22l* was higher in fruit at all fruit developmental stages, except at the 1-cm fruit stage, compared with in other organs. This result suggests that *SlHVA22k* and *SlHVA22l* may actively function throughout the developmental stages of fruit, starting from the cell expansion phase. The mRNA transcript levels of *SlHVA22n* were higher in IM and MG fruit than in fruit at other developmental stages, suggesting its active role in the cell expansion phase of fruit development. *SlHVA22c* is the only gene whose expression peaked in IM fruit, indicating that it may influence the early cell expansion phase of tomato. The higher expression levels of *SlHVA22g* in B and B5 fruits suggested its possible role in tomato fruit ripening. The transcript levels of *SlHVA22i* were high in B fruit (>80-fold higher than in the control) and highest in B5 fruit (~300-fold higher than in the control). This finding indicates that *SlHVA22i* may be a novel gene implicated in the regulation of fruit ripening. The predominant expression of the duplicated gene pair (*SlHVA22a/SlHVA22m*) in the same organ (root) suggested functional conservation, whereas the differential expression levels of other duplicated gene pairs (*SlHVA22e/SlHVA22n* and *SlHVA22g/SlHVA22o*) in different organs suggested functional diversification after gene duplication (Figure 8**)**.

Plant adaptation to diverse environmental stresses is regulated by gene networks, including transcription factors and downstream stress-related genes in ABA-dependent or -independent manners [39,40,41,42]. Previous studies have reported ABA- and stress-induced differential expression of *HVA22* gene family members in various plant species [3,17,18]; interactions between the *cis* elements located in the promoter regions of *HVA22* homologs with several ABA- and stress-related genes; and the exploitation of the *HVA22* promoter as a stress-inducible promoter of stress-related genes in transgenic plants [12,13,14,15,19]. Therefore, it is likely that *SlHVA22* genes function in tomato abiotic stress tolerance.

In the current study, tomato *HVA22* genes displayed differential transcript levels upon exposure to abiotic stress stimuli (Figure 9A–E). Most *SlHVA22* genes were significantly downregulated, while several genes were dramatically upregulated, and only one gene (*SlHVA22j*) was not responsive under cold treatment. This result agrees with a previous study reporting the cold-induced expression of the barley *HVA22* gene and the differential responses of *Arabidopsis HVA22* homologs under cold stress conditions [13,17]. The expression of *SlHVA22b*, *SlHVA22i*, and *SlHVA22n* was highly induced following cold stress (Figure 9A), suggesting their potential role in cold stress tolerance in tomato.

In contrast to cold stress, many *SlHVA22* genes were upregulated, whereas a few genes (*SlHVA22a*, *SlHVA22f*, *SlHVA22h*, and *SlHVA22n*) were downregulated by heat treatment. In response to heat stress, 11 of the 15 *SlHVA22* genes were upregulated, and *SlHVA22i* showed the highest expression (Figure 9B). These genes may be crucial for tomato heat tolerance. These observations are corroborated by a previous study that determined that a mutation in *YOP1*, the yeast (*Saccharomyces cerevisiae*) *HVA22* homolog, results in a growth defect in *yop1* mutants under mild temperature stress (37°C) [5].

Drought treatment upregulated half of the tomato *HVA22* genes and downregulated the other half. *SlHVA22i* had the highest expression level in response to drought stress, followed by *SlHVA22n* (Figure 9C). This suggests that *SlHVA22i* and *SlHVA22n* might actively function in the tomato drought response. These findings are consistent with a previous report in which the expression of *AtHVA22* homologs was differentially regulated by drought and another report in which *CcHVA22d*-overexpressing transgenic tobacco exhibited a lower dehydration rate and buildup of H_2_O_2_ than the WT [6,17].

Under salt stress conditions, the transcript levels of most *SlHVA22* genes increased, but those of a few genes (*SlHVA22d*, *SlHVA22h*, and *SlHVA22o*) decreased. Eleven *SlHVA22* genes were upregulated, with *SlHVA22m* having the greatest expression, followed by *SlHVA22i*, *SlHVA22b*, and *SlHVA22n* (Figure 9D). This indicates the potential involvement of these genes in the salt response of tomato. This is in agreement with previous studies reporting the salt-responsive expression of *HVA22* homologs in yeast and several plant species, and elevated mRNA transcripts of the tomato *HVA22* homolog (*SlHVA22n* in this study) in the salt-tolerant tomato *LeERF1* and *LeERF2* transgenic lines [3,18,20].

ABA is a well-studied phytohormone that regulates a variety of stress-related genes to promote plant tolerance to unfavorable environmental conditions such as cold, heat, drought, and salt stress [43,44,45]. Except for *SlHVA22d*, the expression of all *SlHVA22* genes was altered by ABA treatment, with *SlHVA22i* exhibiting the highest expression, followed by *SlHVA22f* and *SlHVA22n* (Figure 9E). This finding agrees with previous work reporting the responses of *HVA22* homologs from barley and *Arabidopsis* upon exposure to ABA, hinting at their functional role in abiotic stress adaptation in tomato in an ABA-dependent manner [4,5,6,7,8,9,10,11,12,13,14,15,16,17].

We performed a co-expression network analysis of *SlHVA22* genes using RNA sequencing data to further understand their putative functions in tomato. The genes co-expressed with *SlHVA22* genes were involved in diverse biological pathways, including abiotic stress responses and development (Figure 6, Appendix A), suggesting the important biological role of *SlHVA22* genes in tomato. Multiple abiotic stress-responsive genes, such as *SlyHSF-24* (Solyc09g009100), *SlDEAD22* (Solyc07g042010) and *SlDEAD29* (Solyc09g090740) [46,47,48], were co-expressed with *SlHVA22a*, which is in agreement with our finding that *SlHVA22a* responded to abiotic stress treatment. In addition, *SlPIP1;5* (Solyc08g081190), which was highly expressed in roots and under salt treatment, was co-expressed with *SlHVA22a*, whose expression peaked in roots and on exposure to salt stress [49], suggesting that these genes might interact with each other in the root development and salt tolerance of tomato. 

We also identified several genes in the co-expression networks that were related to abiotic stress responses and/or expressed in fruits, such as *SPS1* (Solyc07g007790), *SlPDI7-2* (Solyc11g019920), *SlMC8* (Solyc10g081300), *SlWRKY1* (Solyc07g047960), *SlWRKY3* (Solyc08g081610), Solyc07g064820 (Mitogen-activated protein kinase kinase 2-like), Solyc07g040960 (Salt responsive protein 2), and *SlGPAT6* (Solyc09g014350) in the co-expression network of *SlHVA22g* [50,51,52,53,54,55,56]; *SlRabGAP9a* (Solyc07g049580), *SlRabGAP21a* (Solyc12g009610), *SlGT-33* (Solyc12g043090), *SlFdAT1* (Solyc12g088170), Solyc11g045120 (Translation initiation factor SUI1), and Solyc11g044910 (β-D-xylosidase) in that of *SlHVA22k* [57,58,59,60,61,62,63]; and Solyc10g078600 (phosphate transporter 1-1), *C2H2 zinc finger* (*C2H2-ZF*) (Solyc11g017140), Solyc11g045120 (Translation initiation factor SUI1), Solyc11g044910 (β-D-xylosidase), *SlRabGAP9b* (Solyc12g005930), *SlRabGAP21a* (Solyc12g009610), and *SlFdAT1* (Solyc12g088170) in that of *SlHVA22l* [57,58,59,62,64,65] (Figure 6 and Appendix A). These findings suggest that these three genes, whose transcript levels were high in fruit and induced by abiotic stresses, likely function together with the co-expressed genes in the fruit development and abiotic stress response of tomato. Taken together, the results presented in this study provide useful information for further functional verification of potential *SlHVA22* genes involved in the development and abiotic stress adaption of tomato.

## 4. Materials and Methods

### 4.1. Genome-Wide Identification and Sequence Analysis of SlHVA22 Genes

The *SlHVA22* gene family members were identified from tomato genomes by BLAST searches in the Sol Genomics database (http://solgenomics.net/ accessed on 3 December 2021) [66] using *Arabidopsis thaliana* HVA22 protein sequences retrieved from TAIR (https://www.Arabidopsis.org/ accessed on 3 December 2021) and the HMM profile of SlHVA22 (PF03134) downloaded from the Pfam database (http://pfam.xfam.org/ accessed on 3 December 2021) as query sequences. Subsequently, the NCBI CDD search (https://www.ncbi.nlm.nih.gov/Structure/bwrpsb/bwrpsb.cgi accessed on 4 December 2021) and the SMART web tool (http://smart.emblheidelberg.de/ accessed on 4 December 2021) were used to confirm the presence of the TB2/DP1/HVA22 domain in the resulting fifteen non-redundant HVA22 protein sequences. The transmembrane domains (TMDs) in the SlHVA22 proteins were predicted through the web tool “DeepTMHMM” (https://dtu.biolib.com/DeepTMHMM/ accessed on 20 July 2022) [67]. The ExPASy-ProtParam tool (http://cn.expasy.org/tools/protparam.html accessed on 8 December 2021) was used to analyze the protein length, molecular weight, isoelectric point, and GRAVY values (grand average of hydropathicity index) of each SlHVA22 protein [68]. The open reading frames of the *SlHVA22* genes were determined using the Open Reading Frame Finder tool (https://www.ncbi.nlm.nih.gov/orffinder/ accessed on 8 December 2021). The exon–intron distribution of *SlHVA22* genes was analyzed using the Gene Structure Display Server (GSDS) (http://gsds.cbi.pku.edu.cn/ accessed on 17 December 2021) [69]. A multi-protein sequence alignment was conducted using the Clustal Omega and ESPript web tools [70,71]. The Multiple EM for Motif Elicitation (MEME) online tool (http://meme-suite.org/ accessed on 16 December 2021) was used to determine the conserved motifs in the full-length protein sequences with the following parameters: a maximum number of motifs of 10 and a motif length between six and 50 amino acids [72]. The web server “Immunomedicine Group” (http://imed.med.ucm.es/Tools/sias.html accessed on 14 December 2021) was used to investigate the sequence homology of SlHVA22 proteins. The subcellular localization of the tomato HVA22 proteins was predicted using WoLF-PSORT (https://wolfpsort.hgc.jp/ accessed on 14 December 2021) [73].

### 4.2. Phylogenetic Analysis

The full-length HVA22 protein sequences were aligned with ClustalW and then the phylogenetic tree was built using the neighbor-joining (NJ) algorithm with 1000 bootstrap replications in MEGA 11.0 [74]. The deduced amino acid sequences used in the phylogenetic analysis were retrieved from the Phytozome and NCBI databases, except those of citrus HVA22s, which were obtained from the literature [6]. The names of the genes and their accession numbers used in the phylogenetic tree are described in Appendix A. 

### 4.3. Analysis of Chromosomal Localization, Gene Duplication and Microsyntenic Relationship

The chromosomal locations of the 15 tomato *HVA22* genes identified from the Sol genomic database were used to map the genes to their respective chromosomes with the online tool MapGene2Chrom web v2 (http://mg2c.iask.in/mg2c_v2.0/ accessed on 19 December 2021). The gene duplication events among *SlHVA22* genes were analyzed with the one-step MCScanX algorithm in the TBtools software [75] and examined by BLASTP with an E-value cutoff of 10^−10^. The evolutionary constraint (Ka/Ks) between the tomato duplicated gene pairs of genes was computed using the simple Ka/Ks calculator (NG) in the TBtools software [75]. The divergence time (T) of the duplicated gene pair was predicted using the formula T = Ks/2r Mya (millions of years), where r is the constant for dicotyledonous plants of 1.5 × 10^−8^ substitutions per site per year, and Ks is the synonymous substitution rate per site [76]. The microsyntenic relationship of *HVA22* genes across Arabidopsis, tomato, and rice was explored by performing a reciprocal BLAST search against their whole genomes, and the identified duplicated gene pairs were visualized with the TBtools software [75].

### 4.4. Prediction of Phosphorylation Sites, N-Glycosylation Sites, miRNA Target Sites, and Cis-Regulatory Elements

The putative phosphorylation sites (Ser/Thr/Tyr), N-glycosylation sites (tipo Asn-X-Ser/Thr), and miRNA targets were predicted using the NetPhos 3.1 web-based tool [77], the NetNGlyc 1.0 server (http://www.cbs.dtu.dk/services/NetNGlyc/ accessed on 28 December 2021), and the psRNATarget web tool (http://plantgrn.noble.org/psRNATarget/analysis accessed on 28 December 2021), respectively. The promoter regions of 1500 bp upstream of the initiation codon [ATG] were analyzed to predict the putative *cis*-regulatory elements in SlHVA22 promoters through the PlantCare web server (http://bioinformatics.psb.ugent.be/webtools/plantcare/html/ accessed on 13 December 2021) [78].

### 4.5. 3D Model Prediction of Tomato SlHVA22 Proteins

The comparative modelling of tomato SlHVA22 proteins was performed by the I-TASSER server using the amino acid sequences of SlHVA22a-o as input [49]. The prediction of 3-D models was conducted by multiple threading alignments with LOMETS and iterative structure assembly simulations. The template analogues were identified and the optimal models were chosen based on the highest scores. The ModRefiner was used for the refinement of the modelled structures [79]. The final 3-dimensional models of SlHVA22 proteins were generated using Discovery Studio v.21.1. The putative functions of the resulting modelled proteins were estimated with the I-TASSER server based on global and local homology to template proteins curated in the PDB with identified structures and functions.

### 4.6. Co-Expression Network Analysis of SlHVA22 Genes

Raw RNA-seq reads were downloaded from the SRA database (https://www.ncbi.nlm.nih.gov/sra accessed on 12 December 2021) under the accessions SRR404309, SRR404310, SRR404311, SRR404312, SRR404313, SRR404314, SRR404315, SRR404316, SRR988278, SRR988418, SRR988529, SRR988530-SRR988532, SRR988533-SRR988535, SRR404317-SRR404322, and SRR404324-SRR404329. The FastQC toolkit was used to assess raw sequence reads [80]. The raw reads were cleaned by removing low quality reads and trimming adaptor sequences from the raw data. Then, the cleaned reads were mapped against the tomato reference genome ITAG4.0 using HISAT software and produced a non-redundant genome features annotation file (gff) [81]. Only reads uniquely mapped to the non-redundant gff annotated site were kept for expression analysis. The expression was calculated as the fragment per kb per million reads (FPKM) by using Cuffdiff software. For the co-expression genes with SlHVA22 genes, weighted gene co-expression network analysis (WGCNA) was performed with those genes having FPKM greater than 1 in R [82]. Cytoscape (https://cytoscape.org/ accessed on 30 December 2021) was used to construct the co-expression networks. The GO and KEGG annotations of co-expressed genes with *SlHVA22* genes were conducted by the PANTHER and Kyoto Encyclopedia of Genes and Genomes (KEGG) Server (http://www.genome.jp/kegg/kaas/ accessed on 31 December 2021), respectively. 

### 4.7. Subcellular Localization Analyses

The full-length coding sequences of *SlHVA22* genes were first amplified using the gene-specific primers (Appendix A). Then, they were cloned into the pGA3452 vector to express SlHVA22-GFP fusion proteins under the control of the maize *Ubi1* promoter [83]. The OsAsp1-mRFP fusion protein driven by the 35S promoter served as an ER marker [84]. The SlHVA22-GFP fusion construct and the OsAsp1-mRFP construct were co-transfected into the protoplasts prepared from rice Oc cells through the electroporation method [85]. After the transfected protoplasts were incubated at 28°C in the dark for 12 to 16 h, the localization of the fusion protein was determined using a confocal fluorescence microscope (BX61; Olympus, Tokyo, Japan).

### 4.8. Preparation of Plant Materials and Stress Treatments

Plant materials (seeds of *Solanum lycopersicum* cv Ailsa Craig) were obtained from the Giovannoni laboratory at the Boyce Thompson Institute. Tomato plants were grown in soil in the growth chamber maintained at an adjusted temperature of 25 °C day/20 °C night, a 16 h light/8 h dark photoperiod, a relative humidity of 55 to 70%, and a light intensity of 300 μmol m^−2^s^−1^. For organ-specific expression profiling of *SlHVA22* genes, root, stem, and leaf tissues were harvested from 28-day-old plants. The remaining plants were transferred to a greenhouse adjusted at a temperature of 25 °C/day and 20 °C/night and 65–80% relative humidity for sampling of flowers and fruits at the reproductive stage. Fresh fruits were collected at five different developmental stages: (i) young fruits approximately 7 days after the date of pollination and 1 cm in diameter (1 cm fruits); (ii) immature fruits approximately 21 days after the date of pollination (IM fruits); (iii) mature green fruits approximately 35 days after the date of pollination (MG fruits); (iv) fruits at the breaker stage when the color of mature fruits turns to light yellow-orange from green (B fruits); and (v) fruits 5 days after the breaker stage (B5 fruits). 

Leaf samples from 28-day-old plants with synchronized growth were harvested at 0, 24, 48, 60, and 72 h after the start of the drought treatment and at 0, 1, 3, 9, and 24 h after the start of other abiotic stress treatments, such as cold, heat, salt [NaCl], and abscisic acid [ABA], to examine the expression profiles of *SlHVA22* genes upon exposure to various abiotic stressors. Drought treatment was subjected to the plants by withholding watering for 72 h. The measurement of relative water content (RWC) was conducted in three biological replications using the mature fully expanded leaflet at 0, 24, 48, 60, and 72 h after the drought treatment as described by Collin et al. (2020) (Appendix A) [86]. An ABA treatment was imposed by spraying the leaves of tomato plants with a 100 μM ABA solution. To apply cold and heat stress, the tomato plants were incubated in the growth cabinet maintained at 4 °C and 40 °C for 24 h, respectively. For salt treatment, tomato plants were transferred to the nutrient solution with 200 mM NaCl, and the plants in the nutrient solution were used as the 0 h control [87]. Plants grown in soil under normal conditions (25 °C) were used as the 0 h controls for ABA, heat, cold, and drought treatment. All samples were collected from three independent biological replicates, immediately frozen in liquid nitrogen, and stored at −80 °C for RNA isolation and cDNA synthesis.

### 4.9. RNA Extraction and Quantitative RT-PCR Analysis

Total RNA was extracted from the plant samples with an RNeasy Mini kit (Qiagen, Hilden, Germany) and purified using an RNase-free DNase I kit (Qiagen, Hilden, Germany) as per manufacturer’s instructions. A NanoDrop^®^ 1000 spectrophotometer (Wilmington, DE, USA) was used to measure the quantity and quality of the extracted RNA. cDNA synthesis was performed using 1 µg of total RNA with a Superscript^®^ III First-Strand cDNA synthesis kit (Invitrogen, Carlsbad, CA, USA). The gene-specific primers for all tomato *HVA22* genes were designed using Primer3 software (http://frodo.wi.mit.edu/primer3/input.htm) (Appendix A). The specificity of the amplicons for the primer pairs used in the expression analysis was validated by melting curve analysis [88]. The primer for *Le18S rRNA* (F: AAAAGGTCGACGCGGGCT, R: CGACAGAAGGGACGAGAC) was used as a reference gene for normalization [89]. The reaction mixture for qRT-PCR analysis was composed of 1 µL (50 ng) of cDNA, 2 μL of forward and reverse primers (5 pmol concentration), 2 μL of double distilled water, and 5 µL of iTaq SYBR Green (Qiagen, Hilden, Germany). A Light cycler^®^ 96SW 1.1 instrument (Roche, Germany) was used for amplification and determination of the Cq value of each sample with the following parameters: pre-denaturation at 95 °C for 5 min followed by 40 cycles of 95 °C for 15 s, annealing at 58 °C for 20 s, and extension at 72 °C for 40 s. The 2^−∆∆Ct^ method was used to analyse the relative expression of each gene against each treatment [90]. 

### 4.10. Statistical Analysis

The statistical analysis of the data was performed with SigmaPlot 14 (SYSTAT and MYSTAT Products, United States and Canada) using two-tailed Student’s *t*-tests. * *p* < 0.05, ** *p* < 0.01 and *** *p* < 0.001 were considered statistically significant.

## 5. Conclusions

In the present work, we identified and comprehensively characterized 15 non-redundant *SlHVA22* genes in tomato. We postulate that tomato *HVA22* genes play roles in development processes and abiotic stress responses of plants based on the following evidence: i) the prevalence of hormone- and abiotic stress–responsive *cis* elements, ii) the miRNA target sites and phosphorylation sites in their sequences, iii) their localization to the ER, iv) their predicted transporter activity and binding ability to diverse ligands, v) their co-expression with diverse genes involved in metabolic processes and vi) their differential expression patterns in various organs and under diverse abiotic stresses. The predominant expression of *SlHVA22i* in fruits at the breaker stage and 5 d after the breaker stage revealed that this gene may have an important regulatory role in fruit ripening. Many *SlHVA22* genes, including *SlHVA22b*, *SlHVA22i*, *SlHVA22k*, *SlHVA22l*, *SlHVA22m*, and *SlHVA22n*, were markedly up or downregulated on exposure to various abiotic stressors, hinting that they might be actively involved in the abiotic stress tolerance of tomato. The transcript levels of *SlHVA22b*, *SlHVA22f*, *SlHVA22i*, and *SlHVA22n* were significantly induced by the stress hormone ABA, suggesting that their function in abiotic stress adaption of tomato might be related to ABA. Our results provide valuable information and a solid foundation that may require further functional elucidation of these potential candidate *SlHVA22* genes for the genetic improvement of tomato.

## Figures and Tables

**Figure 1 ijms-23-12222-f001:**
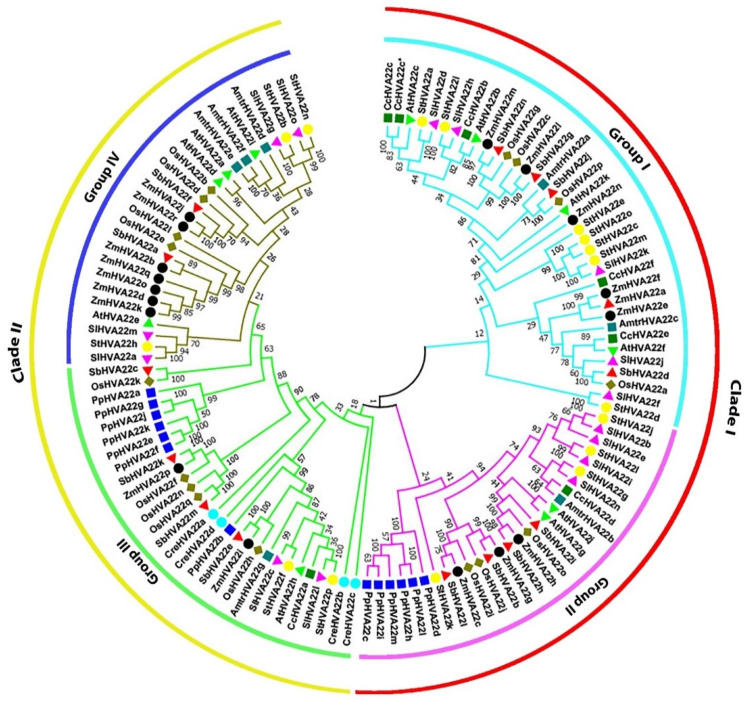
Phylogenetic analysis of HVA22 proteins from tomato and different plant species. The neighbor-joining tree was constructed with the full length HVA22 proteins using ClustalW and MEGA11 with 1000 bootstrap replicates. A species abbreviation was provided prior to each HVA22 protein name: Sl, *Solanum lycopersicum*; St, *Solanum tuberosum;* At, *Arabidopsis thaliana;* Os, *Oryza sativa*; Zm, *Zea mays*; Sb, *Sorghum bicolor*; Amtr, *Amborella trichopoda;* Pp, *Physcomitrella patens*; and *Cre, Chlamydomonas reinhardtii*.

**Figure 2 ijms-23-12222-f002:**
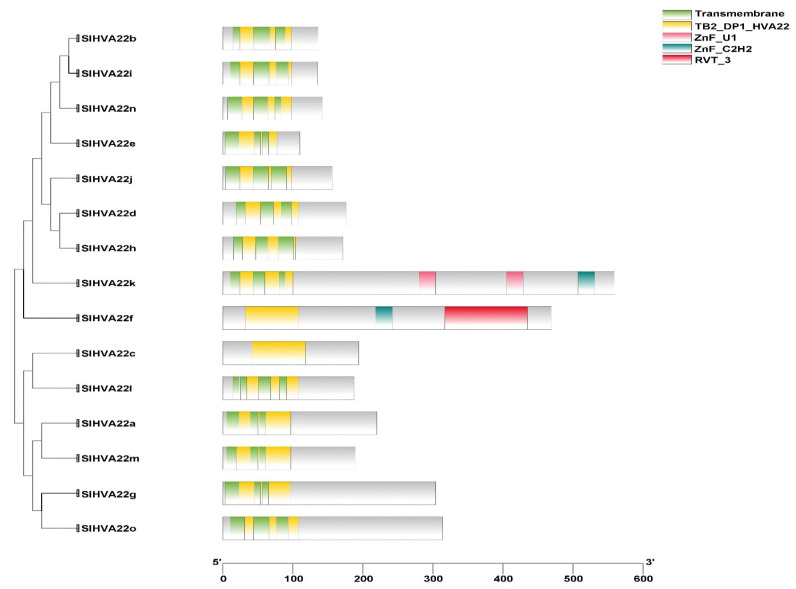
Schematic depiction of the domain organization of SlHVA22 proteins. The transmembrane domain, TB2/DP1/HVA22 domain, U1-type zinc finger (ZnF_U1) domain, the classical C2H2 zinc finger (ZnF_C2H2) domain, and the RVT_3 domain identified in SlHVA22 proteins are shown.

**Figure 3 ijms-23-12222-f003:**
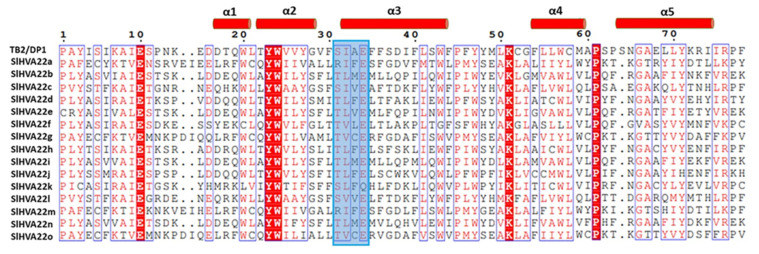
Alignment of TB2/DP1/HVA22 domains of SlHVA22 proteins with that of the typical human TB2/DP1. The secondary structural elements determined by the ESPript 3.0 web tool are indicated above the alignment. The putative conserved casein kinase II sites in the α-helix of the TB2/DP1/HVA22 domain are marked by a blue box.

**Figure 4 ijms-23-12222-f004:**
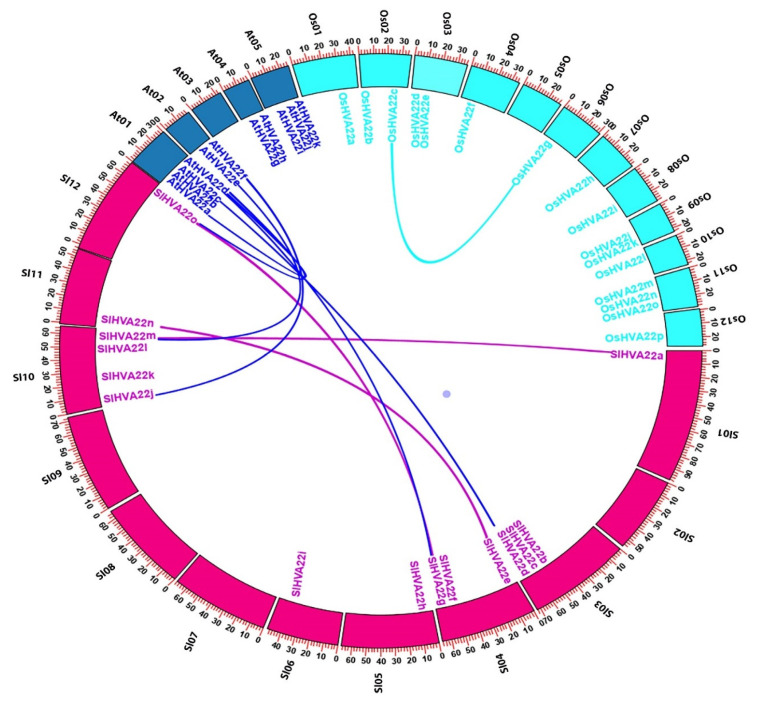
Microsyntenic relationship of *HVA22* genes across Arabidopsis, tomato, and rice. The chromosomes of the three species are depicted by different colors: Arabidopsis, blue; tomato, pink-red; and rice, aqua. All chromosomes are illustrated with the scale in megabase pairs (Mbp). The duplicated *SlHVA22* genes in tomato genomes are linked by pink-red lines.

**Figure 5 ijms-23-12222-f005:**
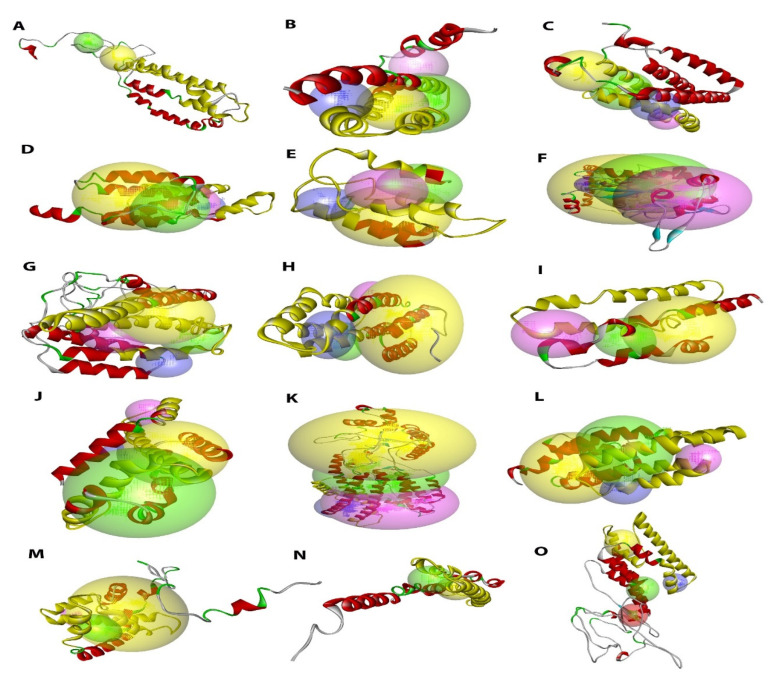
Predicted three-dimensional homology structure of tomato HVA22 proteins. The final 3D structures of SlHVA22 proteins were built by Discovery Studio v.21.1. The secondary structural components: α-helices (red), β-sheets (cyan), coils (green), and loops (gray) as well as the top four putative binding sites: site 1 (yellow sphere), site 2 (green sphere), site 3 (red sphere), and site 4 (blue sphere) are indicated in the predicted 3D models of (**A**) SlHVA22a; (**B**) SlHVA22b; (**C**) SlHVA22c; (**D**) SlHVA22d; (**E**) SlHVA22e; (**F**) SlHVA22f; (**G**) SlHVA22g; (**H**) SlHVA22h; (**I**) SlHVA22i; (**J**) SlHVA22j; (**K**) SlHVA22k; (**L**) SlHVA22l; (**M**) SlHVA22m; (**N**) SlHVA22n; and (**O**) SlHVA22o. The TB2/DP1/HVA22 domain portions in the generated models are underlined in yellow.

**Figure 6 ijms-23-12222-f006:**
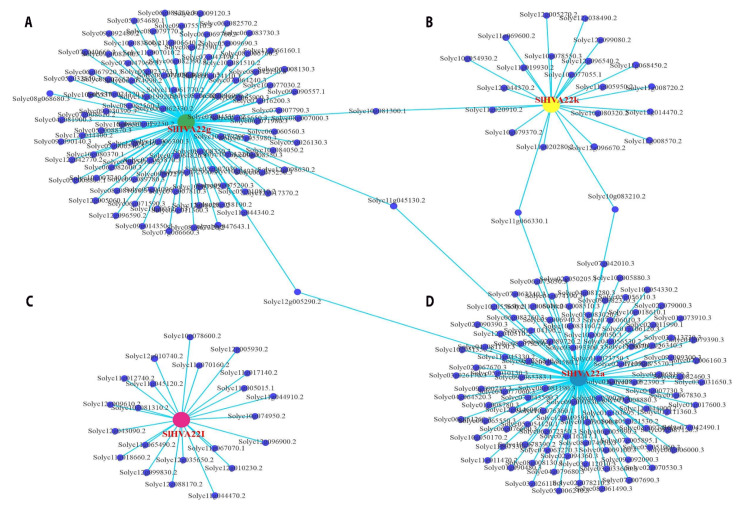
Weighted gene co-expression network analysis (WGCNA) of *SlHVA22* genes. A–D: the co-expressed genes in the network of *SHVA22g* (**A**), *SHVA22k* (**B**), *SlHVA22l*(**C**), and *SlHVA22a* (**D**). The *SlHVA22* genes are marked in red.

**Figure 7 ijms-23-12222-f007:**
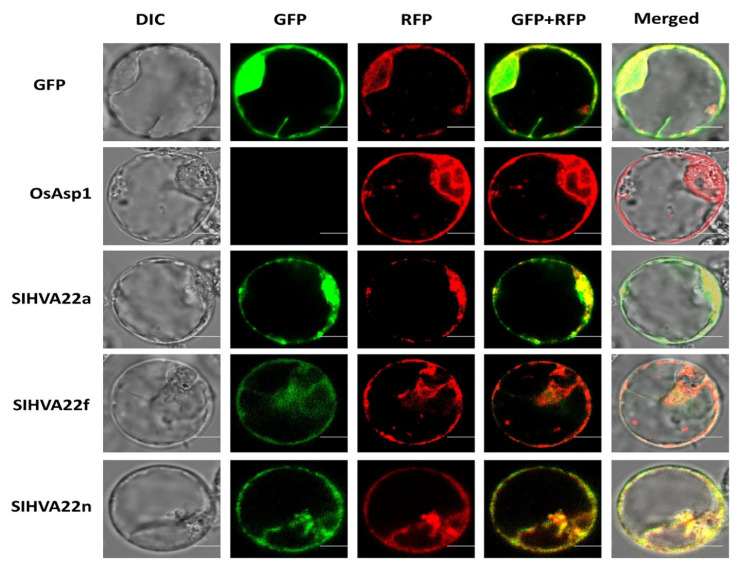
Sub-cellular localization of SlHVA22 proteins. SlHVA22s-SGFP fusion constructs were used to analyze the localization of SlHVA22a, SlHVA22f, and SlHVA22n, and the fluorescence signals were visualized with the confocal microscope. The OsAsp1-mRFP construct was utilized as an ER localization marker. Scale bars = 10 μm.

**Figure 8 ijms-23-12222-f008:**
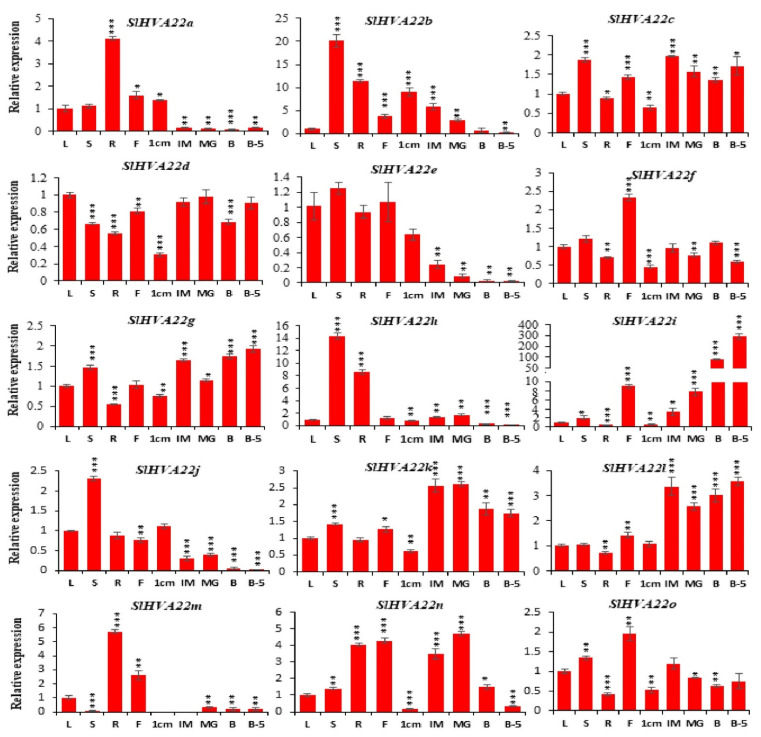
Expression profiles of *SlHVA22* genes in various organs: leaves, roots, stems, flowers, 1 cm fruits, immature fruits (IM), mature green fruits (MG), breaker fruits (B), and fruits 5 days after the breaker stage (B5). The standard deviations of the means of three independent biological replicates are represented by the error bars. The different asterisks above the bars denote the significant variations between the control samples (leaves) and the samples harvested from the other organs, as analyzed by Student’s *t*-test with *p*-values less than 0.05 for *, 0.01 for **, and 0.001 for ***, respectively.

**Figure 9 ijms-23-12222-f009:**
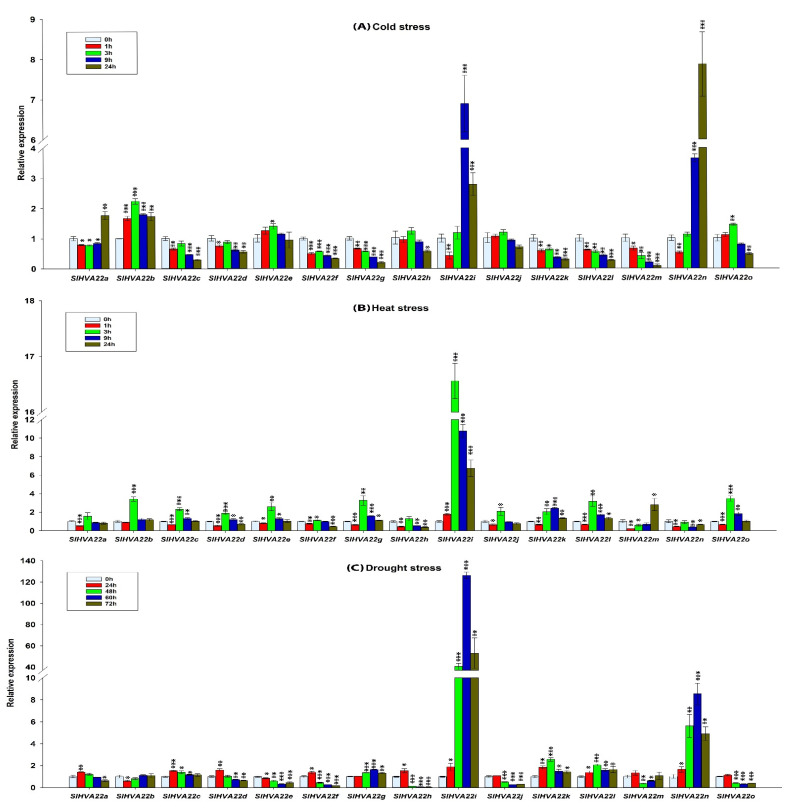
Expression profiling of *SlHVA22* genes under various abiotic stresses: (**A**) cold, (**B**) heat, (**C**) drought, (**D**) salt, and (**E**) ABA treatment. Error bars represent standard deviations of the means of three independent biological replicates of qRT-PCR analysis. The various asterisk marks (* for *p*-value < 0.05, ** for *p*-value < 0.01, and *** for *p*-value < 0.001) determined using the student *t*-test indicate the statistically significant differences between the control samples (0 h) and treated samples of *HVA22* genes.

**Table 1 ijms-23-12222-t001:** Predicted Ka/Ks ratio of the duplicated *SlHVA22* gene pair along with its divergence time.

Duplicated Gene Pair			Ka	Ks	Ka/Ks	Duplication Type	Types of Selection	Time (MYA)
*SlHVA22a*	vs.	*SlHVA22m*	0.256048578	0.755904898	0.338731206	Segmental	Purifying selection	25.20
*SlHVA22e*	vs.	*SlHVA22n*	0.190039202	0.413123219	0.460006102	Segmental	Purifying selection	13.77
*SlHVA22g*	vs.	*SlHVA22o*	0.183462269	0.580908552	0.315819535	Segmental	Purifying selection	19.36

Ks, the number of synonymous substitutions per synonymous site; Ka, the number of non-synonymous substitutions per nonsynonymous site; MYA, million years ago.

## Data Availability

Not applicable.

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
