# Peer review of "Comprehensive Genome-Wide Analysis and Expression Pattern Profiling of the SlHVA22 Gene Family Unravels Their Likely Involvement in the Abiotic Stress Adaptation of Tomato"

_ijms, 2022, doi:10.3390/ijms232012222_

Round 1

Reviewer 1 Report

In this study, authors have described identification and characterization of 15 members of  HVA22 gene family in tomato. Authors have performed comprehensive gene expression analysis via RT-qPCR in various tissues and under stress/hormone conditions in addition to the sub-cellular location experiments. Overall, the analyses presented is of sufficient quality, though no functional analysis of candidate gene/s is presented. There are few comments given below for consideration to improve the quality.

- The details given in the first section of Results is completely unnecessary and can be restricted to only one line with reference to the Table (Table can also be moved to Suppl. material). First three section of the results can be combined in one and presented together with substantially reduced text.

- It is not clear if all the RT-PCR reactions were performed in biological replicates. The analyses of three independent biological replicates is must for reliable results.

- It is not clear that on what basis the three genes were selected for experimental analyses of sub-cellular location.

- Figure 7 is unnecessary and can be omitted or moved to Suppl. material.

- Discussion part is very shallow. Most of the content in the present version is repetition of the results. It needs to be revised comprehensively to clearly reflect the discussion and significance of the results obtained in view of the current literature.

Reviewer 2 Report

In this study, Antt Htet Wai et al. reported the identification and characterization of HVA22 gene family in tomato. In total 15 non-redundant SlHVA22 genes were identified. Co-expression analysis indicated that these genes may be responsive to stress stimuli. Moreover, comprehensive RT-qPCR were performed to demonstrate that subset of HVA22 family genes were able to response to various abiotic stimuli. This study is well designed and conducted to high standard. Data was solid and shown in a logic way. I only have minor concerns regarding some missing controls in stress treatment experiments, as well as the figure panel label.

1.     Authors had performed various abiotic stress treatments, the molecular effects of each treatment should be validated aside from HVA22 gene family. The reason why I brought this issue is that abiotic stimuli is highly variable and sensitive to many factors, including chemical dose, duration and so on. My suggestion is that, for each treatment, authors should select 1-2 established marker genes for qPCR and show that the abiotic stress efficiently triggered the activation of related marker genes. This will make the changes in HVA22 genes more convincible.  

2.     Since there are multiple images in figure 9 and 10, images in different panels should be labeled as A, B and so on. In addition, the label system should be consistent across figures, and lower case (a,b) in Figure 10 should be changed to upper case A, B…

3.      All the font and size of panel labels should be consistent and meet the criteria of IJMS. For example, font of panel label (A) in figure 5 is different from that in figure 6. Size of Label of y-axis in figure 7 is different from that in figure 9, 10. Authors should go through the manuscript and correct all the mistakes seriously.  
